# Relative quantification of BCL2 mRNA for diagnostic usage needs stable uncontrolled genes as reference

Nehanjali Dwivedi[1,2☯], Sreejeta Mondal[1☯], Smitha P. K.[1☯], Sowmya T.[1], Kartik Sachdeva[1], Christopher Bathula[1], Vishnupriyan K.[1], Nataraj K. S.[3], Sharat Damodar[3], Sujan K. Dhar[4]*, Manjula Das[1,4]*

**1** Tumor Immunology Program, MSMF, MSMC, Narayana Health City, Bangalore, India, **2** MAHE, Manipal, India, **3** Department of Haematology, MSMF, MSMC, Narayana Health City, Bangalore, India, **4** Beyond Antibody, InCite Labs, MSMF, MSMC, Narayana Health City, Bangalore, India

☯ These authors contributed equally to this work.
* manjula.das@ms-mf.org (MD); sujan@beyondantibody.com (SKD)

**Data Availability Statement:** All relevant data are within the manuscript and its Supporting Information files.

## Abstract

Dysregulation of BCL2 is a pathophysiology observed in haematological malignancies. For implementation of available treatment-options it is preferred to know the relative quantification of BCL2 mRNA with appropriate reference genes. For the choice of reference genes—(i) Reference Genes were selected by assessing variation of >60,000 genes from 4 RNA-seq datasets of haematological malignancies followed by filtering based on their GO biological process annotations and proximity of their chromosomal locations to known disease translocations. Selected genes were experimentally validated across various haematological malignancy samples followed by stability comparison using geNorm, NormFinder, BestKeeper and RefFinder. (ii) 43 commonly used Reference Genes were obtained from literature through extensive systematic review. Levels of BCL2 mRNA was assessed by qPCR normalized either by novel reference genes from this study or GAPDH, the most cited reference gene in literature and compared. The analysis showed PTCD2, PPP1R3B and FBXW9 to be the most unregulated genes across lymph-nodes, bone marrow and PBMC samples unlike the Reference Genes used in literature. BCL2 mRNA level shows a consistent higher expression in haematological malignancy patients when normalized by these novel Reference Genes as opposed to GAPDH, the most cited Reference Gene. These reference genes should also be applicable in qPCR platforms using Taqman probes and other model systems including cell lines and rodent models. Absence of sample from healthy-normal individual in diagnostic cases call for careful selection of Reference Genes for relative quantification of a biomarker by qPCR.BCL2 can be used as molecular diagnostics only if normalized with a set of reference genes with stable yet low levels of expression across different types of haematological malignancies.

## Introduction

Overexpression of BCL2 (B-cell lymphoma 2), a mitochondrial membrane protein has been observed in several haematological malignancies due to genetic and epigenetic mechanisms

**Funding:** The study is funded by GLUE grant-scheme number BT/PR23078/MED/29/1253/2017 by Department of Biotechnology (http://dbtindia.gov.in/), Govt of India, awarded to SD and MD. The funders had no role in study design, data collection and analysis, decision to publish, or preparation of the manuscript.

**Competing interests:** The authors have declared that no competing interests exist.

resulting in evasion of apoptosis, giving the malignant cells a longer life span and survival benefits at times of nutrient deficiency, hypoxia and growth factor deprivation [1–3]. Estimation of level of BCL2 along with other anti-apoptotic genes are essential to avail efficient treatment-options by R-CHOP (Regimen of Cyclophosphamide, Doxorubicin, Vincristine and Prednisone and Rituximab) or Venetoclax in different haematological malignancies [4, 5]. By visualization of chromosomal aberrations using karyotyping or FISH (Fluorescence in-situ Hybridization), BCL2 levels can be inferred indirectly [6]. Detection of expression of BCL2 protein by immunohistochemistry, a standard pathological testing procedure for DLBCL, has not been adopted in the clinics for bone marrow tissues of liquid cancers due to sample inconsistency and challenging procedure of capturing low concentrations of biomarkers [7]. Western blotting, for the very nature of the method, cannot be adopted for high throughput pathological testing. ELISA, for detection of BCL2 in human plasma remains limited since only one splice isoform of the mitochondrial membrane protein is available in soluble form, thus bringing down the effectiveness of the assay [8]. BCL2 at the mRNA level can be determined without ambiguity by next generation sequencing [9], NanoString [10] and microarray [11], though increasing time and expense of pathological testing. In clinical trials relative quantification by qPCR (Quantitative Polymerase Chain Reaction)can be successfully used due to the availability of appropriate controls in untreated or normal groups [12, 13]. Although being time and cost-effective, it suffers misinterpretation in pathological setting since the relative quantification depends only on the RG (Reference Gene) used due to the absence of normal samples.

Normalization with a RG which shows varying expression across samples can often lead to wrong conclusions as seen with the use of Glyceraldehyde-3-Phosphate Dehydrogenase (GAPDH) as RG in gene expression studies of pulmonary tuberculosis [14] and CD8+ T-cells under inactivated or activated condition [15]. Similarly, ABL proto-oncogene 1 (ABL1), the recommended RG for gene expression studies with leukemic patients [16] was found to have extremely low expression in neutrophils [17] making it unsuitable as RG for the specific case. Such discrepancies have prompted researchers to analyze gene expression across multiple tissues [18] or pan-cancer database like TCGA [19] to propose normalization factors using multiple RG candidates.

This study, through a systematic review of literature in haematological malignancies concluded that mostly conventionally used "house-keeping" genes are still being deployed (S1 Table and S1 Fig) despite their varied expression based on cell type, developmental stage and experimental conditions [20] with rare exceptions [21, 22]. None of the 43 genes thus identified could be used to relatively quantify BCL2 as molecular diagnostics since compared to the FPKM (Fragments Per Kilobase of transcript per Million mapped reads) value of the anti-apoptotic genes across 4 databases (S2 Fig), most of the RGs from the literature are not only higher, but also varied significantly (S3 and S4 Figs) with few exceptions. Inspired by genome wide search for RGs from publicly available RNA-seq or microarray data in human and other organisms [23–25], we report here a set of novel candidate RGs obtained from an unbiased search of >60,000 genes in haematological malignancies to be used to normalize BCL2 and other anti-apoptotic genes in qPCR as molecular diagnostics.

## Materials and methods

### Ethics statement

The study was performed in compliance with ethical practices and was approved by Narayana Health Academics Ethics Committee, Narayana Health Hospitals (Ethics Approval Number: NHH/AEC-CL-2017-152A).

## Systematic review of commonly used RGs

Literature search was carried out in PubMed database(PubMed) [26] as detailed in S5 Fig according to PRISMA (Preferred Reporting Items for Systematic Reviews and Meta-Analyses) guidelines [27].

**Selection of stable genes.** Protein-coding genes identified from 4 publicly available datasets (Table 1), using ensembldb annotation package within R statistical software, were categorised into four quartiles based on their median expression values across all samples. Genes with median expression in middle two quartiles (Q2 and Q3) in all datasets were considered, as Q1 and Q4 representing extreme ends of the expression spectrum are not preferred as RG candidates for normalization of molecular diagnostic markers.

To determine the stability of a gene, following statistical measures were employed–(i) CV = $\bar{x}/\sigma_x$ where $\bar{x}$ and $\sigma_x$ are mean and standard deviation of a variable $x$ respectively and (ii) normality p-value as measured by Shapiro-Wilks test, where a p-value less than 0.05 signifies that the distribution is away from Normal [28]. CV, although used most frequently isn't a robust measure as it is affected by outliers. To solve this, a third parameter was used: MAD (Median Absolute Deviation) = $median|x - \hat{x}|$ where $\hat{x}$ is the median of $x$, after normalization with median [29]. MAD is a better measure for understanding the spread of the distribution as it depends on medians, a parameter, less prone to deviations by outliers.

Low or comparable statistical variation across samples (represented by low values of CV and MAD) and a normal distribution (high value of normality p-value or low values of 1 –p-value) are characteristics of an ideal RG. Therefore genes with median expression values in middle quartiles (Q2 and Q3) were shortlisted and clustered based on their CV, MAD and 1 – p-value (normalized to their respective z-scores) using PAM (Partitioning around Medoids) algorithm [30]. Required optimal number of clusters was calculated using Silhouette graphical method [31]. For each tissue sample, the gene cluster with the lowest medoid value of parameters was selected and the genes at the intersection of the four clusters were shortlisted [24].

The list was further filtered by analysing and eliminating genes based on stop words in their GO (Gene Ontology) annotation such as transcription factors, nuclear receptor or other nuclear localization, DNA binding activity, response to external stimuli, translational and transcriptional activation, since genes with such characteristics regulated by environmental conditions are unsuitable as RG candidates. Next, genes were ranked in ascending order of their mean Euclidean distance $d = \sqrt{CV^2 + MAD^2 + (1 - p)^2}$ (all parameters replaced by their z-scores) in this three-parameter hyperspace for each dataset. Average of $d$ across four datasets was taken to calculate the mean Euclidean distance ($\bar{d}$). Genes with $\bar{d}$ < median were selected for further

**Table 1. List of RNA-seq databases.**

| Dataset | Disease | Tissue | Samples (n) | Source | Download Location |
|---------|---------|--------|-------------|--------|-------------------|
| TCGA-LAML | AML | Blood | 151 | TCGA Research Network | https://www.cancer.gov/tcga |
| TARGET-AML | Paediatric AML | Bone marrow | 159[*] | | |
| GDC-DLBC# | DLBCL | Lymph nodes | 562 | Schmitz et al 2018[49] | https://gdc.cancer.gov/about-data/publications/DLBCL-2018 |
| MMRF-MM | Multiple Myeloma | Bone marrow | 779[†] | Multiple Myeloma Research Foundation | https://research.themmrf.org |

[*] Both primary and recurrent tumor

[†] only 1st visit records.

# FPKM data for GDC-DLBC dataset was available as log2 transformed normalized value which was converted to FPKM.

analysis. Locus of genes associated with pathogenic translocations were identified [32, 33] and candidate RGs in close proximity of such loci (within 10 bands in the same arm of chromosome) were eliminated by an automated method. Further, only genes with non-zero FPKM value in all samples from four datasets were retained. Then, each gene was given a composite quartile ranking (CQR, the sum of quartile indices from each dataset) and genes with CQR value {8, 9, 10} (median expression in $2^{nd}$ quartile in at least two datasets) were shortlisted (S6 Fig).

## Design of primers

**BCL2 primers.** BCL2 has two known splice isoforms: membrane-bound BCL2α and a less studied soluble BCL2β, lacking the trans-membrane domain at the 3' C-terminal [34]. Most reported primers amplified only BCL2α or larger amplicon (S2 Table), hence new primers were designed (Table 2).

**RG primers.** Primers for shortlisted genes were designed (Table 2; S3 Table) using PrimerBank [35] and IDT [36].

## Sample details

RNA was isolated from peripheral blood or bone marrow samples from patient or normal individuals (S7 Fig) with their informed consent (Ethics Approval Number: NHH/AEC-CL-

**Table 2. Primers details of RGs and BCL2.**

| Primer | Accession No | Sequence (5' - 3') | Amplicon Length (bp) | Tm (°C) | Amplification Factor |
|--------|--------------|--------------------|----------------------|---------|----------------------|
| ACY1 | NM_000666 | Fw 5'-CACTGACAACCGCTATATCCG | 106 | 60.3 | 2.05 |
| | | Rv 5'-CTCATGCAGCCGTTCATCGT | | 62.9 | |
| ANKRD26 | NM_014915 | Fw 5'-TCTCGGCAAGATCCACAAAGC | 119 | 62.7 | 2.06 |
| | | Rv 5'-AATGTAGAGCCGTCCTGTTCA | | 60.9 | |
| JMJD4 | NM_001161465 | Fw 5'-GTCTGTCAATGTCTGTGGGAG | 127 | 60 | 1.99 |
| | | Rv 5'-CAGGTGTGTGTCGCAGAGT-3' | | 61.9 | |
| PTCD2 | NM_024754.5 | Fw 5'-TATGGGACACTGCACATCAC-3' | 118 | 62 | 1.99 |
| | | Rv 5'-GGCTGACCATCCTCTTGTTTA-3' | | 62 | |
| PPP1R3B | NM_024607 | Fw 5'-AGAACCTCGCATTTGAGAAGAC-3' | 109 | 60.3 | 1.93 |
| | | Rv 5'-TCTGAACCGGCATAAGTGTCC-3' | | 61.8 | |
| FBXW9 | NM_032301 | Fw 5'-TAGGGCGGTGCGATGATTC-3' | 117 | 61.9 | 1.99 |
| | | Rv 5'-CGGATTTTGGCGGACTGAGA-3' | | 62.2 | |
| NANP | NM_152667.3 | Fw 5'-GGTCCGCCTACTTCTATTAACG-3' | 112 | 62 | 1.98 |
| | | Rv 5'-TCTCTGCTCTCCACCTACAA-3' | | 62 | |
| PLEKHM3 | NM_001080475.3 | Fw 5'-GATGATATCAGCCCAGCCTTAG-3' | 109 | 62 | 1.94 |
| | | Rv 5'-GGACTTCCTGGATCCCATAAAC-3' | | 62 | |
| TSGA10 | NM_025244 | Fw 5'-TACTCAGCGACACCTTGCTAA-3' | 152 | 60.9 | 2 |
| | | Rv 5'-CCAGATCATTGAGGGTTCCAC-3' | | 60.1 | |
| NAT1 | NM_001160174 | Fw 5'-GGGAGGGTATGTTTACAGCAC-3' | 128 | 60.1 | 1.8 |
| | | Rv 5'-ACATCTGGTATGAGCGTCCAA-3' | | 60.9 | |
| RIC8B | NM_018157 | Fw 5'-ATAGTGTTCAACAGTCAGATGGC-3' | 133 | 60.3 | 1.92 |
| | | Rv 5'-GCAAGCGCAAGTCAAAGCA-3' | | 62.2 | |
| GAPDH | NM_001289745.3 | Fw 5'-TCGACAGTCAGCCGCATCTTCTTT-3' | 196 | 59 | 1.95 |
| | | Rv 5'-GCCCAATACGACCAAATCCGTTGA-3' | | 60 | |
| BCL2 | NM_000657.2 | Fw 5'-GGAGGATTGTGGCCTTCTTT-3' | 113 | 62 | 1.89 |
| | | Rv 5'-GCCCAATACGACCAAATCCGTTGA-3' | | 60 | |

Fw: Forward primer; Rv: Reverse primer.

2017-152A). Subjects with Hepatitis B/C or HIV, and pregnant or lactating women were excluded from the study.

PBMC/BMMC (Peripheral Blood Mononuclear cells/ Bone Marrow mononuclear cells) were separated by layering 1:1 of blood or bone marrow (diluted to 1:3 with 1X PBS Gibco™, Germany) above Ficoll-Paque Plus Histopaque, (HiMedia, India) followed by centrifugation at 400 RCF for 25 Mins with brakes off. Resultant buffy coat was washed twice with 1X PBS and once with 1X PenStrep (HiMedia, India) before culturing at cell density of 0.5 to 0.8 million cells/mL of RPMI 1640 (HiMedia, India) with 20% FBS (Gibco™, Germany, Brazil origin) and 1X PenStrep for subculturing the lymphocyte population.

## RNA, cDNA and qPCR

From FFPE (Formalin-fixed, paraffin-embedded) blocks, 5–8 curls were deparaffinized in xylene at 50˚C, followed by proteinase K (HiMedia, India) treatment prior to RNA isolation. Either from lymphocytes or from deparaffinized retrospective samples [37] RNA was isolated by TRIzol™ (Ambion, US) method and quantified with Qubit RNA BR Assay Kit (Thermo Fisher Scientific, US) before converting to cDNA using SuperScript IV (SSIV, Thermo Fisher Scientific, US) as per manufacturers' instructions. With no-template control (NTC), qPCR was performed in triplicates for each sample using KAPA SyBr green Universal reagents (Sigma Aldrich, US), cDNA (1:10 dilution) and primers in a 5μL reaction mix (qPCR condition: pre-incubation at 95˚C for 10 minutes followed by amplification for 40 cycles–denaturation at 95˚C for 10 sec, amplification at 60˚C for 15 sec, and extension at 72˚C for 15 sec) in Roche LightCycler 480 II machine.

## Optimization of primers

Primers were optimized for qPCR as required by the MIQE guidelines [38]. All primers were used at four different final concentrations (forward/reverse): 200nM/200nM; 200nM/100nM; 100nM/200nM and 100nM/100nM with pooled cDNA template, obtained from six normal healthy volunteers to yield single amplification product. Primer efficiency was checked using a two-fold five-point dilution of the template. Primer efficiency was obtained from standard curve using the formula: $Amplication\ Factor = \left(10^{-\frac{1}{slope}} - 1\right) \times 100$ (Table 2).

## Stability analysis of candidate RGs

Mean of Cq (Quantification cycle) of NTC were subtracted from Cq values of each gene in qPCR experiments to obtain ΔCq = $Cq\ (sample) - Mean\ Cq\ (NTC)$ and relative expression as $E^{-\Delta Cq}$ for each replicate, where E is the amplification factor of corresponding gene.

Stability of expression of the candidate RGs was analysed using three independent algorithms–geNorm [39], NormFinder [40] and BestKeeper [41] and the web-based RefFinder tool [42] that integrates all three algorithms plus the delta CT method. Algorithm geNorm was run using the SLqPCR R package [43], whereas author-supplied R package and Excel worksheet were used for NormFinder and BestKeeper analysis respectively. Mean Cq values for each gene for all 78 samples were used as input for BestKeeper and RefFinder, whereas for geNorm and NormFinder relative expression values were used. Since NormFinder uses a model-based approach to quantify inter- and intra-group variations, the malignant and non-neoplastic or healthy-normal samples were used as two groups for NormFinder analysis.

Comprehensive stability rank of each gene was calculated as the geometric mean of stability rank given by each method.

## Expression analysis of BCL2

RQ (Relative Quantification) of BCL2 expression was calculated either as ratio of relative expression of BCL2 with relative expression of GAPDH or the normalization factor which is geometric mean of relative expression of three candidate RGs:

$$RQ\,(GAPDH) = E^{-\Delta Cq}(BCL2)/E^{-\Delta Cq}(GAPDH)$$

$$RQ\,(proposed) = E^{-\Delta Cq}(BCL2)/Geo\,Mean\,E^{-\Delta Cq}(PTCD2, PPP1R3B, FBXW9)$$

## Results and discussion

Quantification by qPCR could be the choice of pathology laboratories for a quick and cost-effective platform for single-gene expression level, with appropriate RG. Towards this effort, MacRae et al (2013) [21] performed a genome wide search and statistical analysis using RNA-seq data from 55 leukemia patients. In a more recent pan-cancer study [22], publicly available gene expression data from 193 microarray studies were analysed to identify a few RG candidates that showed minimal variation between malignant and normal samples and were validated in droplet digital PCR on bone marrow samples of ALL patients. We have used 6 types of haematological malignancy samples encompassing bone marrow, PBMC and FFPE blocks along with non-neoplastic bone marrow and healthy PBMC samples subsequent to using much wider publicly available data from 1,651 samples in AML, DLBCL and multiple myeloma databases. Further, we have employed an improved statistical analysis including clustering technique described in Methods section. Instead of an ad hoc approach of selection of top few genes from the clusters, we used important biological considerations to further prune the list of candidate RGs.

## Systematic review of commonly used RGs from literature

Systematic Review of 122 articles yielded 43 RGs used in haematological malignancies through (a) selection of genes by different analysis methods (S4 Table) and (b) usage of known RGs in qPCR (S1 Table). FPKM values of all these RGs when examined in 4 public databases showed varied expression among different types of haematological malignancies (S3 and S4 Figs) with maybe the exception of PGGT1B [22]. However, since other genes selected in the literature showed higher expression and correlated extreme variation, we could not depend on the assay and proceeded to select novel RGs with an unbiased approach.

## Selection of candidate RGs

**Statistical analysis.** Stepwise filtration of the number of genes from each dataset is summarized in S6 Fig and also in Graphical Abstract. Fig 1 shows gene clusters plotted in CV, normalized MAD and 1-p-value hyperspace for four datasets. Cluster marked in green in each figure represents the cluster with least medoid value (S5 Table) for the three parameters. Selected clusters in the four datasets had an overlap of 1961 genes, indicating large number of genes involved in housekeeping processes and hence showing lesser inter-sample variation across diverse datasets. Common genes were pruned further to 541 by GO biological process term filtration, disease association and CQR to lead to a final of 19 genes (S6 Table) that were taken through experimental validation. Melt curve analysis and efficiency check with pooled cDNA from six healthy volunteers narrowed it down to 11 genes with stable median expression and single amplification product of expected size for each (Table 2). Primers for genes which did not qualify the efficiency check were eliminated as they failed to show single amplification peak after repeated trials with new experimental conditions and even new primer sequences (S3 Table).

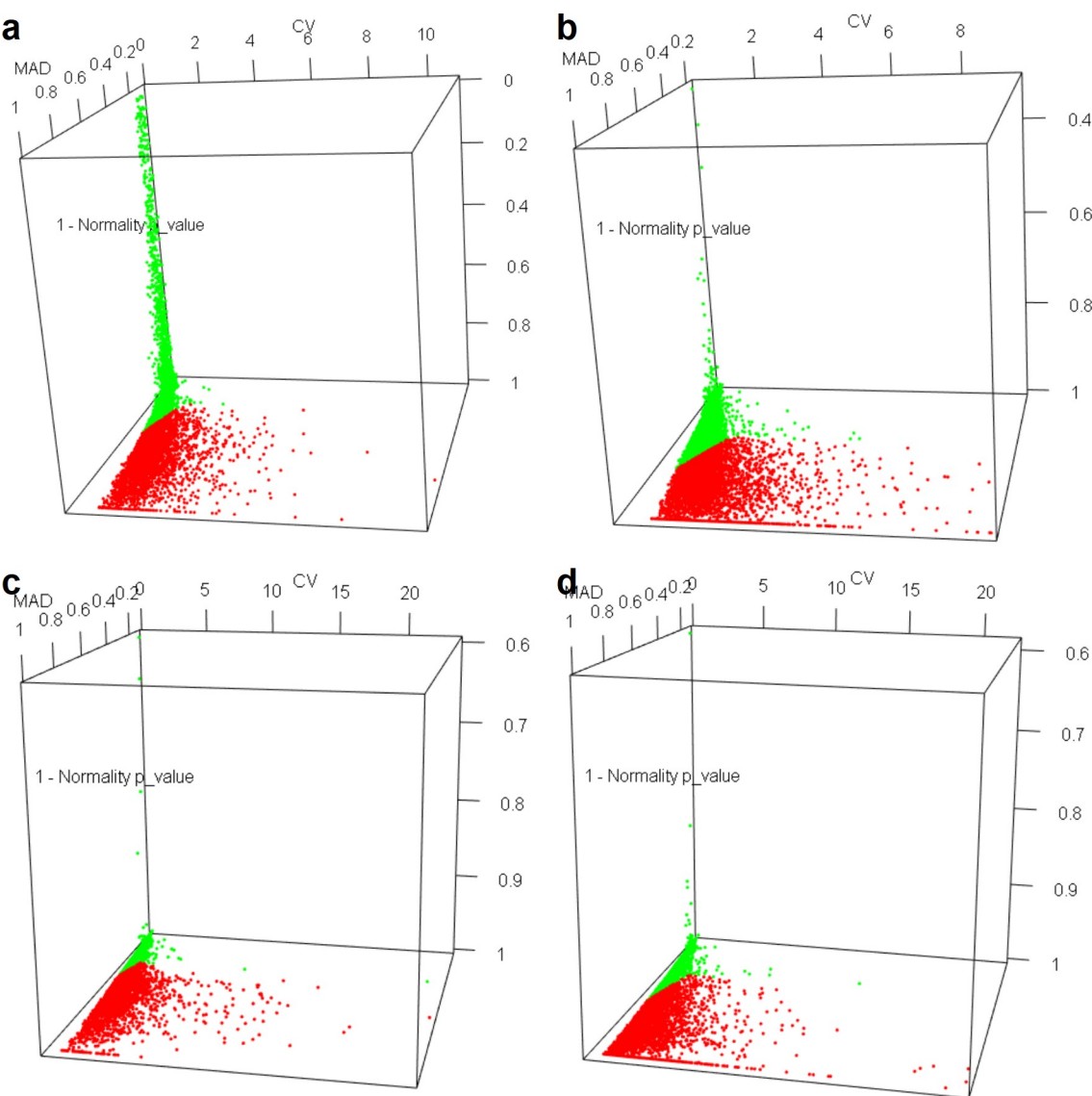

**Fig 1. Statistical analysis of candidate genes.** Genes plotted in the CV, normalized MAD and (1 –p-value) hyperspace for the four datasets (A) TCGA-LAML, (B) TARGET-AML, (C) GDC-DLBC and (D) MMRF-MM. Cluster shown in green represents the chosen cluster with least value of medoids.

Expression of 11 genes with efficient primers were analysed on 78 samples by qPCR. Using observed Cq values preliminary stability analysis of the 11 genes were done with online RefFinder tool to select top 10 stable genes (PTCD2, PPP1R3B, FBXW9, NANP, RIC8B, JMJD4, PLEKHM3, NAT1, ANKRD26, TSGA10) as RG candidatessss.

**Stability analysis of candidate RGs.** Results of BestKeeper algorithm, used independently or as part of RefFinder were comparable whereas results of geNorm or Normfinder analysis differed as they used different inputs. Geometric mean of stability ranks assigned in each algorithm was used to create comprehensive stability ranking of all the candidate RGs (S7 Table and Fig 2). The analysis shows PTCD2, PPP1R3B and FBXW9, to be most stable across all analysed patient samples.

PTCD2 (Pentatricopeptide repeat-containing protein 2) codes for a mitochondrial protein, involved in RNA binding, maturation and respiratory chain function though its exact

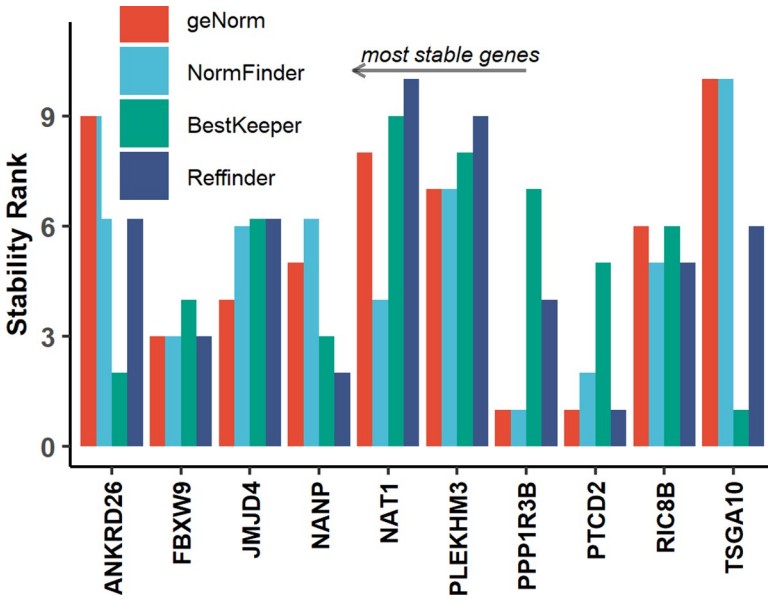

**Fig 2. Stability rank of candidate reference genes.**

molecular function is not well understood [44, 45]. PPP1R3B (Protein phosphatase-1 regulatory-subunit-3B) encodes for a catalytic subunit phosphatase 1 regulatory subunit 3B, which is involved in hepatic glycogen dysregulation in Type 2 diabetes [46–48] FBXW9 (F-box/WD repeat-containing protein 9) is a cytosolic protein involved in ubiquitination and proteasome degradation [49].

## Expression analysis of BCL2

Accurate determination of BCL2 expression among few anti-apoptotic markers in patients with haematological malignancies is emerging as a critical diagnostic test for clinicians to suggest efficacious therapy options. FPKM values of 8 RGs (5 common and 3 novel) from the publicly available databases when compared (Fig 3) with BCL2 indicated the novel RGs to be better normalization candidate for BCL2 in qPCR assays in pathology labs due to less and stable expression.

Comparison of relative expression of GAPDH versus the proposed normalization factor (geometric mean of relative expression of the three RG candidates) clearly show a large variation in GAPDH expression ($0–2^{15}$) across malignant samples (Fig 4A; S8 Table). Granted its popularity, the expression stability of GAPDH has been proven to differ in different conditions due to its involvement in apoptotic cell death through ubiquitin ligase [50], membrane trafficking [51], upregulation in AML [52], involvement in Non-Hodgkin's B-cell Lymphomas [53] and inconsistency in several other cancers [54]. On the other hand, proposed RGs have lesser variation ($0–2^{10}$) and their expressions are consorted with each other making them better candidate as RG compared to GAPDH. This behaviour is translated to BCL2 expression RQ in malignant samples when normalized with GAPDH (Fig 4B). Evidently, normalization with GAPDH underestimates relative quantification of BCL2 compared to normalization with proposed RGs, with a statistically significant difference in median values (p < 0.01, Wilcoxon rank sum test) between the two schemes. BCL2 quantification in haematological malignancies by qPCR is overtly reliant on RG since availability of "adjacent normal" sample is ruled out. Above results clearly demonstrate how the quantification may go off limit due to a wrong choice of RG.

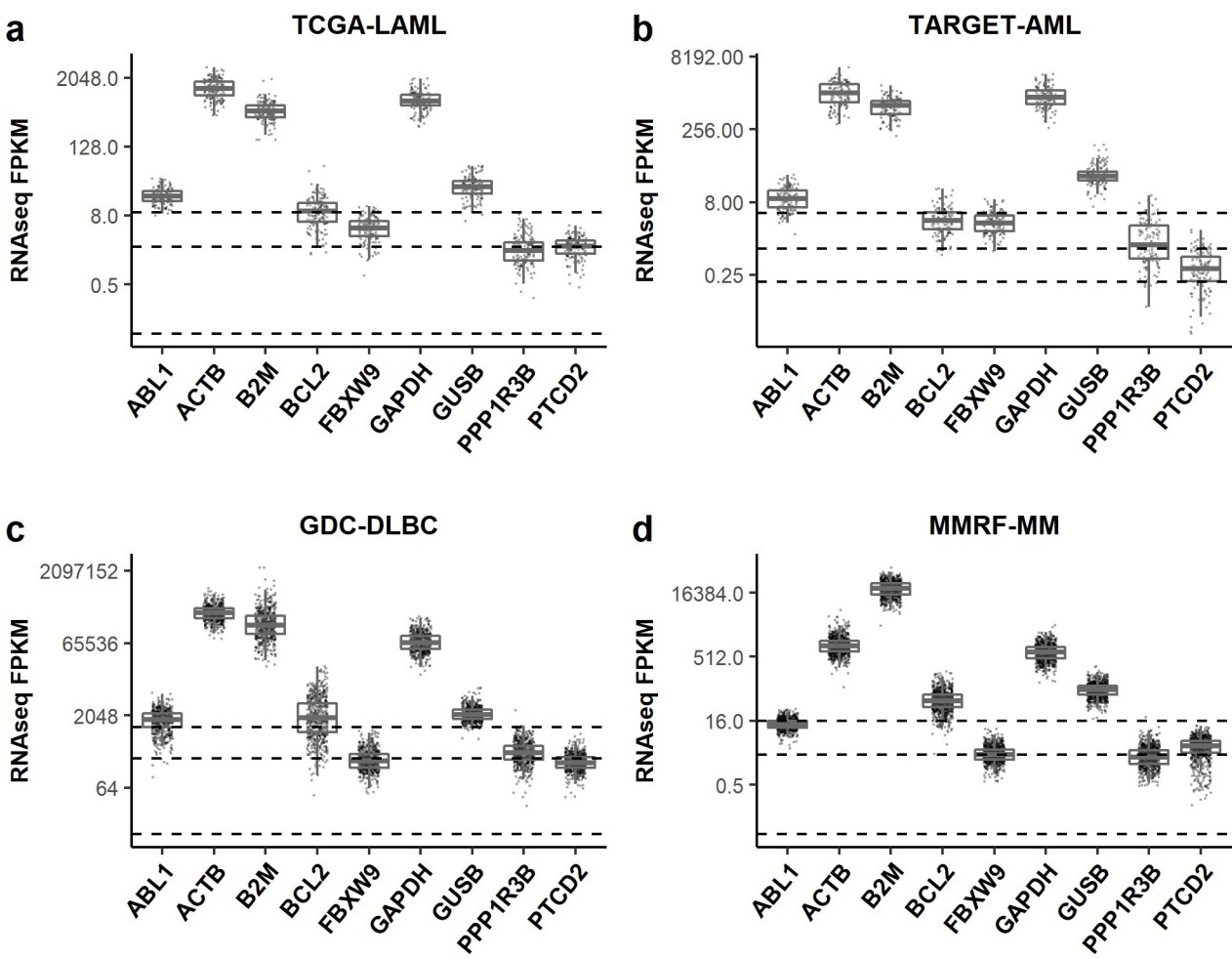

**Fig 3. Candidate reference genes in hematological malignancy datasets.** Expression values of candidate genes in four datasets (A) TCGA-LAML, (B) TARGET-AML, (C) GDC-DLBC and (D) MMRF-MM.

## Broader applicability of proposed reference genes

Though primary objective of this study is to discover RG candidates for BCL-2 diagnostics in a clinical setting, the RGs may have broader utility in other experimental platforms or model systems. In the systematic review we found a number of research articles [55–62] that have used TaqMan probes instead of SYBR green, whereas our validation experiment was carried out using SYBR green probes. However, studies in different contexts such as a tropical oilseed plant [63], or measurement of expression of various adenosine receptors in breast cancer tissue [64] and in experiments using human reference RNA [65], SYBR green PCR assays were observed having fair concordance with TaqMan PCR. From these evidences, we believe that stability of proposed RGs is not likely to differ between SYBR green and TaqMan qPCR assays.

To assess variation of these stable RGs in cell lines, we analyzed RPKM values of 18,778 protein-coding genes across 173 cell lines of haematopoietic and lymphoid tissue origin from Broad Institute Cancer Cell Encyclopedia [66] and found the proposed RGs presenting much lesser variations in expression compared to the 5 common RGs (GAPDH, ABL1, B2M, GUSB and ACTB) in cell lines as well (S8 Fig).

Both transgenic and wild type and occasional rat models are widely used in leukemia and lymphoma research [67, 68]. Usability of RGs common between clinical and animal studies

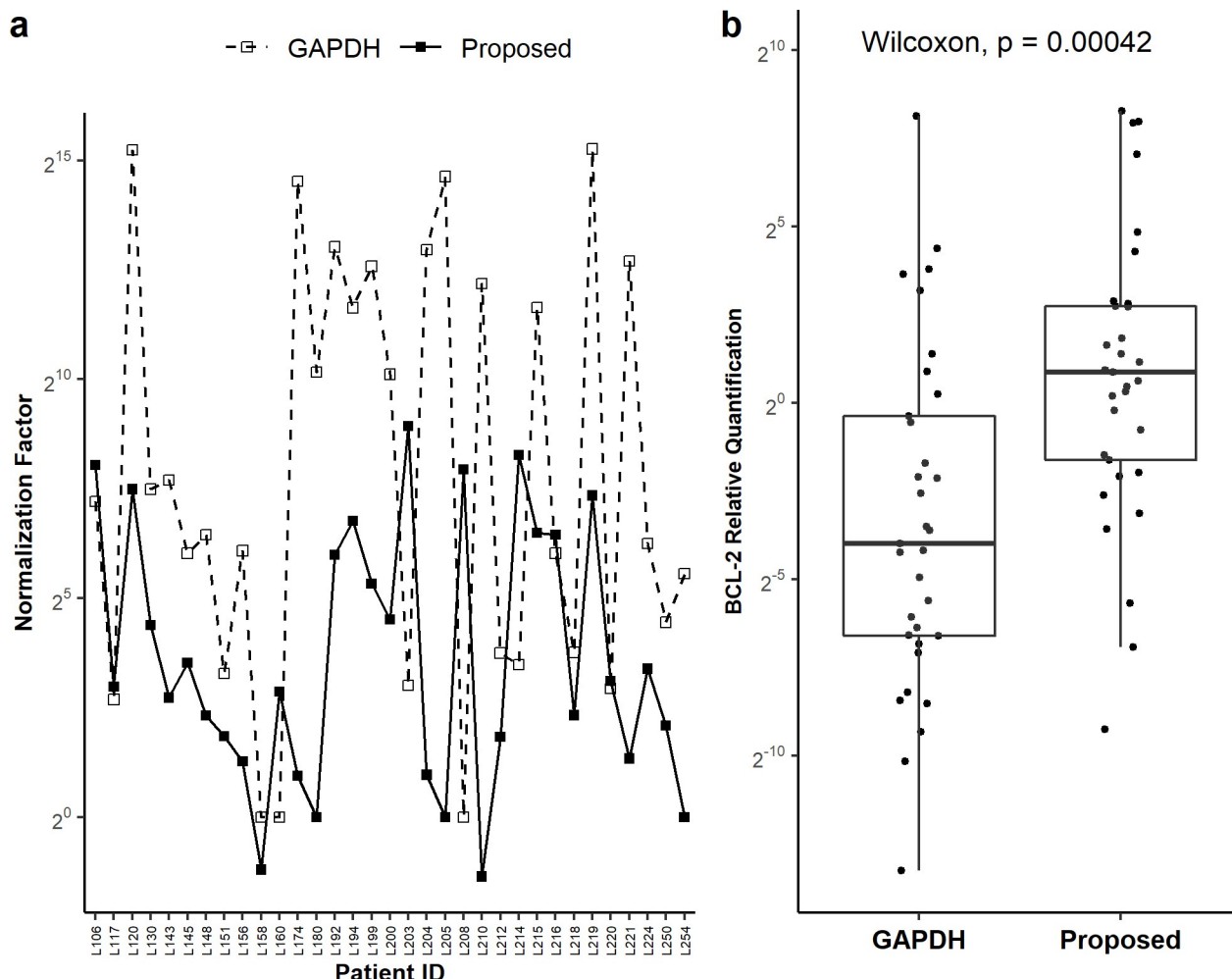

**Fig 4. Relative expression of chosen reference genes and relative quantification of BCL2.** (A) Relative expression of chosen reference genes (solid lines) and GAPDH (dashed line) across patient samples. (B) Relative Quantitation of BCL2 expression with respect to the candidate reference genes and GAPDH in malignant patient samples.

will thus be of immense advantage. We find that the proposed RGs–PTCD2, PPP1R3B and FBXW9 have 70–90% sequence similarity and identity with corresponding genes in mice and other commonly used rodent models (S9 Table), suggesting the genes playing similar role in cellular function thereby displaying stability, similar to that in humans. Hence normalization factor derived from the expression of these RGs may be applicable in murine and other rodent models as well with suitable design of primers encompassing conserved regions.

Beyond detection of gene expression at mRNA level, it may be worthwhile to explore the applicability of protein counterpart of the stable RGs in Western Blot as control for protein detection. By design, we have chosen RGs that are of moderate expression level (in middle quartiles of expression among other genes), and they may not be detectable by Western blot unless a larger amount of sample is loaded, which is often not feasible with clinical samples. However, it may be an interesting proposition to predict stable reference proteins for use in Western blot by statistical analysis of proteomics data and associated systematic review of literature.

## Conclusion

Our results indicate that genes PTCD2, PPP1R3B and FBXW9 render more reliability to qPCR-based diagnostic test of BCL2 in haematological malignances. The conclusion can be extended to other biomarkers in liquid cancer as well as for research with other model systems such as cell lines and rodents.

## Supporting information

**S1 Table. List of reference genes in literature.**
(DOCX)

**S2 Table. List of BCL2 primers from literature.**
(DOCX)

**S3 Table. List of unqualified primers.**
(DOCX)

**S4 Table. Literature explaining analysis and selection of reference gene.**
(DOCX)

**S5 Table. Z-score Medoid values.**
(DOCX)

**S6 Table. List of selected genes.**
(DOCX)

**S7 Table. Individual and combined stability rank and scores of candidate reference genes.**
(DOCX)

**S8 Table. Relative expression of GAPDH and the proposed normalization factor.**
(DOCX)

**S9 Table. Sequence similarity and identity with corresponding genes in mice, rat and guinea pig.**
(DOCX)

**S1 Fig. RGs found in literature with more than one citation.**
(TIFF)

**S2 Fig. FPKM values of BCL2 family of anti-apoptotic genes in the four datasets.**
(TIFF)

**S3 Fig. FPKM values of RGs found in relevant literature with more than one citation.**
(TIFF)

**S4 Fig. FPKM values of RGs found in relevant literature with a single citation.**
(TIFF)

**S5 Fig. Workflow according to PRISMA guidelines for systematic Review for commonly used reference genes.**
(TIFF)

**S6 Fig. Statistical analysis workflow.**
(TIFF)

**S7 Fig. Patient samples used in the study.**
(TIFF)

**S8 Fig. Variation in stable RGs in cell lines and animal model.**
(TIFF)

**S1 Graphical Abstract.**
(TIFF)

## Acknowledgments

Authors acknowledge Prof. Joy Kuri, Chair, Department of Electronic Science and Engineering, Indian Institute of Science, Bangalore for providing the computational resources.

## Author Contributions

**Conceptualization:** Sujan K. Dhar, Manjula Das.

**Data curation:** Nehanjali Dwivedi, Sreejeta Mondal, Smitha P. K., Sowmya T., Kartik Sachdeva, Christopher Bathula, Vishnupriyan K.

**Formal analysis:** Sujan K. Dhar.

**Funding acquisition:** Sharat Damodar, Manjula Das.

**Investigation:** Nehanjali Dwivedi, Sreejeta Mondal, Smitha P. K., Sowmya T.

**Methodology:** Nehanjali Dwivedi, Sreejeta Mondal, Smitha P. K., Sowmya T., Vishnupriyan K., Manjula Das.

**Project administration:** Manjula Das.

**Resources:** Nataraj K. S., Sharat Damodar.

**Software:** Sujan K. Dhar.

**Supervision:** Manjula Das.

**Validation:** Nehanjali Dwivedi, Sreejeta Mondal, Smitha P. K., Sowmya T., Kartik Sachdeva, Christopher Bathula, Vishnupriyan K.

**Visualization:** Manjula Das.

**Writing – original draft:** Sreejeta Mondal, Sujan K. Dhar.

**Writing – review & editing:** Nehanjali Dwivedi, Sreejeta Mondal, Smitha P. K., Sujan K. Dhar, Manjula Das.

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
