## [Decision Letter · Decision Letter 0]

12 Jun 2020

PONE-D-20-15589

Relative quantification of BCL2 mRNA for diagnostic usage needs stable uncontrolled genes as reference

PLOS ONE

Dear Dr. das,

Thank you for submitting your manuscript to PLOS ONE. After careful consideration, we feel that it has merit but does not fully meet PLOS ONE’s publication criteria as it currently stands. Therefore, we invite you to submit a revised version of the manuscript that addresses the points raised during the review process.

Please make sure you address issues relating to inter-gene normalization and, considering that expression levels depend seriously on the choice of reference gene, authors must address the need for normalization of these parameters. Namely, the inter-gene normalization approach for reference gene - at best with support of data retrieved from comparing these reference genes.

We look forward to receiving your revised manuscript.

Kind regards,

Pedro V. Baptista, PhD

Academic Editor

PLOS ONE

Journal Requirements:

Additional Editor Comments (if provided):

Authors should carefully provide response to the Reviewers' queries. Particularly, attention must be paid to the normalization factor and how to apply the same rationale to other relevant genes involved in the condition. Careful observation of the inter-gene variations and how to reduce impact in variation in quantifying gene expression must be taken into account.

Reviewers' comments:

Reviewer's Responses to Questions

**Comments to the Author**

1. Is the manuscript technically sound, and do the data support the conclusions?

Reviewer #1: Yes

Reviewer #2: Yes

2. Has the statistical analysis been performed appropriately and rigorously? 

Reviewer #1: Yes

Reviewer #2: Yes

3. Have the authors made all data underlying the findings in their manuscript fully available?

Reviewer #1: Yes

Reviewer #2: Yes

4. Is the manuscript presented in an intelligible fashion and written in standard English?

Reviewer #1: Yes

Reviewer #2: Yes

5. Review Comments to the Author

Reviewer #1: The manuscript submitted by Dwivedi et al screened out three stable uncontrolled reference genes for qPCR analysis of BLC2 mRNA. The authors validated the significance and applicability of PTCD2, PPP1R3B and FBXW9 across lymph-nodes, bone marrow and PBMC samples after massive statistic analysis. The methodology established here demonstrates a compelling and professional approach to addressing the issues in quantifying expression of BCL2 levels of hematological malignancies in the clinics. Therefore, the results presented here indicated much advancement in this field and would be of great interest to a large number of readers. I would support the publication of this well-written manuscript in PLOS One after minor revision.

1. Would these screened out reference genes be stable in cell lines, other than primary cells?

2. Have the authors compared or noticed any difference between Taqman and SyBr green in qPCR when determining the stability of reference genes?

3. Would the western blot assay reflect similar results using these three stable reference genes?

4. How about the applicability of these three reference genes in animal model or murine cells?

Reviewer #2: Dysregulation of the B-cell leukemia/lymphoma-2 (BCL-2) family of proteins of the intrinsic apoptotic pathway is fundamental to the pathophysiology of many hematologic malignancies. BCL-2 dysregulation enables cells that would normally undergo apoptosis to survive. Targeting BCL-2 has been the goal of various initiatives, and some are in clinical trials and for example Venetoclax was approved by the US Food and Drug Administration for the treatment of patients with CLL who have 17p deletion. This article sought to assess the reliability of several Reference Genes (RG) in the quantification of BCL-2 expression in haematological malignancies. RGs are indispensable for normalizing mRNA levels across samples in real-time quantitative PCR. As their expression levels vary under different experimental conditions, appropriate reference gene selection is thus critical for gene-expression studies. As such the work has merit.

The authors found a set of novel candidate RGs obtained from an unbiased search of >60,000 genes in haematological malignancies to normalize BCL2 and other anti-apoptotic genes in qPCR. The workflow was thorough, and included looking at publicly available datasets and most importantly, experimental validation in a series of patient samples. The analysis shows PTCD2, PPP1R3B and FBXW9 to be most stable genes across all patient samples.

Reference to past publications underlining the importance of a correct selection of appropriate RGs is needed. The use of an inappropriate reference gene may lead to erroneous normalization and flawed conclusions. Given that no universal reference gene exists with a constant expression in all tissues and cell types, reference genes need to be selected and validated for each specific application. The use of more than one reference gene and even gene panels has been recommended to overcome variations in individual gene expression. This should be stressed in the article. Examples of articles are given below.

Radonić, A., Thulke, S., Mackay, I. M., Landt, O., Siegert, W., & Nitsche, A. (2004). Guideline to reference gene selection for quantitative real-time PCR. Biochemical and biophysical research communications, 313(4), 856-862.

Dheda, K., Huggett, J. F., Chang, J. S., Kim, L. U., Bustin, S. A., Johnson, M. A., ... & Zumla, A. (2005). The implications of using an inappropriate reference gene for real-time reverse transcription PCR data normalization. Analytical biochemistry, 344(1), 141-143.

As discussed by Vandesompele et al., in order to measure expression levels accurately, normalization by multiple housekeeping genes instead of one is required. Consequently, a normalization factor based on the expression levels of the best-performing housekeeping genes must be calculated. This has not been done in this article, therefore the authors are urged to present normalization factors with the RGs they identified and present the results.

6. PLOS authors have the option to publish the peer review history of their article (what does this mean?). If published, this will include your full peer review and any attached files.

Reviewer #1: No

Reviewer #2: No

---

## [Author Response · Author response to Decision Letter 0]

20 Jun 2020

Response to Reviewer’s comment on “Relative quantification of BCL2 mRNA for diagnostic usage needs stable uncontrolled genes as reference” by Dwivedi et al submitted to PLOS ONE

Responses to Reviewer 1 comments

R1.1. Would these screened out reference genes be stable in cell lines, other than primary cells?

Response: We thank the reviewer for bringing up this aspect. We checked expression of these reference genes in cell lines of haematopoietic and lymphoid tissue origin from Broad Institute Cancer Cell Encyclopaedia and found them to show lesser variations compared to common reference genes (GAPDH, ABL1, B2M, GUSB and ACTB) in cell lines as well. These results are added in revised manuscript (Results and Discussions: lines 262-268) and in supplementary data (Fig S8).

R1.2. Have the authors compared or noticed any difference between Taqman and SyBr green in qPCR when determining the stability of reference genes? 

Response: We have not performed any experiment comparing stability of the reference genes between TaqMan and SYBR green probes. While performing the systematic review we came across a number of research articles [1–8] that have used TaqMan probes, but none of them present a comparison with SYBR green. 

However, we came across other studies that have compared the assay using TaqMan and SYBR green probes. Cao and Shockey [9] demonstrated identical expression patterns of reference genes between SYBR Green and TaqMan qPCR for a tropical oilseed plant whereas Tajadini et al [10] measured the expression of various adenosine receptors in breast cancer tissue and found performance of the SYBR green assay as comparable with TaqMan. In a different context using human reference RNA, Arikawa et al [11] found SYBR green PCR assays having maximum concordance with TaqMan PCR. 

From these evidences, we believe that stability of reference genes are not likely to differ in SYBR green qPCR assay compared to TaqMan. Hence, we have not commented on this aspect in our revised manuscript.

R1.3. Would the western blot assay reflect similar results using these three stable reference genes?

Response: Our study focuses on detection of BCL2 through qPCR for diagnostic purposes, hence aiming at only the mRNA expression of genes. 

By design we have chosen reference genes that are of moderate expression level (in middle quartiles of expression among other genes), and they may not be detectable by Western blot unless a larger amount of sample is loaded, which is often not feasible with clinical samples. 

Having said that, it may be an interesting proposition to predict stable reference proteins for use in Western blot. For this a new study can be initiated with statistical analysis of proteomics data and associated systematic review of literature. 

R1.4. How about the applicability of these three reference genes in animal model or murine cells?

Response: This a very relevant comment as use of animal models, especially murine models are very common in leukemia and lymphoma research. We analyzed sequences of the proposed reference genes and found them to have 70 – 90% similarity and identity with corresponding genes in mouse or other rodent models. This result indicates that proposed reference genes may also be useful in experiments with murine models. 

We have incorporated these results in revised manuscript (Results and discussion: lines 269 – 277) and in supplementary data (Table S8). We thank the reviewer for bringing out this perspective.

Responses to Reviewer 2 Comments

R2.1. Reference to past publications underlining the importance of a correct selection of appropriate RGs is needed. 

Response: We had mentioned about inconsistency in GAPDH expression as reported in literature in the Results and discussion section (lines 303 – 306). However, as suggested by reviewer we have added a paragraph on this topic (Introduction: lines 64-72) adding reference to past publications. We thank the reviewer for this valuable suggestion.

R2.2. The authors are urged to present normalization factors with the RGs they identified and present the results.

Response: Normalization factor as mentioned by reviewer refers to geometric mean of relative expression of the reference genes. We had already used this in our calculations for BCL2 relative quantitation.

However, based on reviewer’s suggestion we have explicitly mentioned the normalization factor in appropriate places of the manuscript (Materials and methods: lines 199-204, Results and discussion: lines 300-301 and Fig. 4). We have also added a table with values of normalization factors in the supplementary data (Table S9). 

References 

1. Barbany G, Hagberg A, Olsson-Strömberg U, Simonsson B, Syvänen AC, Landegren U. Manifold-assisted reverse transcription-PCR with real-time detection for measurement of the BCR-ABL fusion transcript in chronic myeloid leukemia patients. Clin Chem [Internet]. 2000 Jul [cited 2020 Jan 10];46(7):913–20. 

2. Bolufer P, Sanz GF, Barragán E, Sanz MA, Cervera J, Lerma E, et al. Rapid quantitative detection of BCR-ABL transcripts in chronic myeloid leukemia patients by real-time reverse transcriptase polymerase-chain reaction using fluorescently labeled probes. Haematologica [Internet]. 2000 Dec 1 [cited 2020 Jun 16];85(12):1248–54. 

3. Scholl C, Breitinger H, Schlenk RF, Döhner H, Fröhling S, Döhner K. Development of a real-time RT-PCR assay for the quantification of the most frequent MLL/AF9 fusion types resulting from translocation t(9;11)(p22;q23) in acute myeloid leukemia. Genes Chromosom Cancer. 2003 Nov 1;38(3):274–80. 

4. Abruzzo L V., Lee KY, Fuller A, Silverman A, Keating MJ, Medeiros LJ, et al. Validation of oligonucleotide microarray data using microfluidic low-density arrays: A new statistical method to normalize real-time RT-PCR data. Biotechniques. 2005;38(5):785–92. 

5. Vera-Lozada G, Scholl V, Barros MHM, Sisti D, Guescini M, Stocchi V, et al. Analysis of biological and technical variability in gene expression assays from formalin-fixed paraffin-embedded classical Hodgkin lymphomas. Exp Mol Pathol. 2014 Dec 1;97(3):433–9. 

6. Lantuejoul S, Rouquette I, Blons H, Stang N Le, Ilie M, Begueret H, et al. French multicentric validation of ALK rearrangement diagnostic in 547 lung adenocarcinomas. Eur Respir J. 2015 Jul 1;46(1):207–18. 

7. Spiess B, Rinaldetti S, Naumann N, Galuschek N, Kossak-Roth U, Wuchter P, et al. Diagnostic performance of the molecular BCR-ABL1 monitoring system may impact on inclusion of CML patients in stopping trials. PLoS One. 2019 Mar 1;14(3). 

8. Casoli C, Pilotti E, Bertazzoni U. Proviral load determination of HTLV-1 and HTLV-2 in patients’ peripheral blood mononuclear cells by real-time PCR. Methods Mol Biol. 2014;1087:315–23. 

9. Cao H, Shockey JM. Comparison of TaqMan and SYBR green qPCR methods for quantitative gene expression in tung tree tissues. J Agric Food Chem. 2012 Dec 19;60(50):12296–303. 

10. Tajadini M, Panjehpour M, Javanmard S. Comparison of SYBR Green and TaqMan methods in quantitative real-time polymerase chain reaction analysis of four adenosine receptor subtypes. Adv Biomed Res. 2014;3(1):85. 

11. Arikawa E, Sun Y, Wang J, Zhou Q, Ning B, Dial SL, et al. Cross-platform comparison of SYBR® Green real-time PCR with TaqMan PCR, microarrays and other gene expression measurement technologies evaluated in the MicroArray Quality Control (MAQC) study. BMC Genomics [Internet]. 2008 Jul 11 [cited 2020 Jun 17];9(1):328. Accessed: 10.1186/1471-2164-9-328

---

## [Decision Letter · Decision Letter 1]

26 Jun 2020

PONE-D-20-15589R1

Relative quantification of BCL2 mRNA for diagnostic usage needs stable uncontrolled genes as reference

PLOS ONE

Dear Dr. das,

Thank you for submitting your manuscript to PLOS ONE. After careful consideration, we feel that it has merit but does not fully meet PLOS ONE’s publication criteria as it currently stands. Therefore, we invite you to submit a revised version of the manuscript that addresses the points raised during the review process.

Authors are requested to introduce into the manuscript two minor discussion points that were provided in the response to the Referees

1- the brief clarification on SYBR green (the discussion text in the Response letter should suffice)

2- the point on the focus on mRNA expression and referring the correlation to protein expression (as measured by Western blot) - again, the discussion text in the Response letter should suffice.

We look forward to receiving your revised manuscript.

Kind regards,

Pedro V. Baptista, PhD

Academic Editor

PLOS ONE

Additional Editor Comments (if provided):

Authors are requested to introduce two minor discussion points that were provided in the response to the Referees

1- the brief clarification on SYBR green (the discussion text in the Response letter should suffice)

2- the point on the focus on mRNA expression and referring the correlation to protein expression (as measured by Western blot) - again, the discussion text in the Response letter should suffice.

Reviewers' comments:

Reviewer's Responses to Questions

**Comments to the Author**

1. If the authors have adequately addressed your comments raised in a previous round of review and you feel that this manuscript is now acceptable for publication, you may indicate that here to bypass the “Comments to the Author” section, enter your conflict of interest statement in the “Confidential to Editor” section, and submit your "Accept" recommendation.

Reviewer #1: All comments have been addressed

Reviewer #2: All comments have been addressed

2. Is the manuscript technically sound, and do the data support the conclusions?

Reviewer #1: Yes

Reviewer #2: Yes

3. Has the statistical analysis been performed appropriately and rigorously? 

Reviewer #1: Yes

Reviewer #2: Yes

4. Have the authors made all data underlying the findings in their manuscript fully available?

Reviewer #1: Yes

Reviewer #2: Yes

5. Is the manuscript presented in an intelligible fashion and written in standard English?

Reviewer #1: Yes

Reviewer #2: Yes

6. Review Comments to the Author

Reviewer #1: The authors have greatly revised the manuscript and improved its overall qualify. My questions have been well addressed. I would recommend it for publication in PLOS one.

Reviewer #2: (No Response)

7. PLOS authors have the option to publish the peer review history of their article (what does this mean?). If published, this will include your full peer review and any attached files.

Reviewer #1: No

Reviewer #2: No

---

## [Author Response · Author response to Decision Letter 1]

1 Jul 2020

Response to Editor’s comments on “Relative quantification of BCL2 mRNA for diagnostic usage needs stable uncontrolled genes as reference” by Dwivedi et al submitted to PLOS ONE

Editor’s comments: 

Authors are requested to introduce into the manuscript two minor discussion points that were provided in the response to the Referees

1- the brief clarification on SYBR green (the discussion text in the Response letter should suffice)

2- the point on the focus on mRNA expression and referring the correlation to protein expression (as measured by Western blot) - again, the discussion text in the Response letter should suffice.

Response:

We thank Editor for the suggestion to introduce above discussion points in the manuscript which has further elevated the value of our work. In line with this suggestion, we have introduced a new subsection “Broader applicability of proposed reference genes” under Results and Discussion and included discussion points on SYBR green (lines 310 – 319), cell lines (lines 320 - 324), rodent models (lines 325- 333) and Western blot (lines 334 - 341) in this subsection. We think all these discussion points being under same subsection makes the flow more coherent.

---

## [Editor Report · Decision Letter 2]

7 Jul 2020

Relative quantification of BCL2 mRNA for diagnostic usage needs stable uncontrolled genes as reference

PONE-D-20-15589R2

Dear Dr. das,

We’re pleased to inform you that your manuscript has been judged scientifically suitable for publication and will be formally accepted for publication once it meets all outstanding technical requirements.

Kind regards,

Pedro V. Baptista, PhD

Academic Editor

PLOS ONE

---

## [Editor Report · Acceptance letter]

13 Jul 2020

PONE-D-20-15589R2 

Relative quantification of BCL2 mRNA for diagnostic usage needs stable uncontrolled genes as reference 

Dear Dr. Das:

I'm pleased to inform you that your manuscript has been deemed suitable for publication in PLOS ONE. Congratulations! Your manuscript is now with our production department. 

Kind regards, 

on behalf of

Prof. Pedro V. Baptista 

Academic Editor

PLOS ONE